# Open Charm Measurements at CERN SPS Energies with the New Vertex Detector of the NA61/SHINE Experiment: Status and Plans [†]

**Anastasia Merzlaya [1,2,*] and on behalf of the NA61/SHINE Collaboration**

[1]  Faculty of Physics, Astronomy and Applied Computer Science, Jagiellonian University, 31-007 Krakow, Poland
[2]  Faculty of Physics, Saint Petersburg State University, 199034 Saint Petersburg, Russia
[*]  Correspondence: anastasia.merzlaya@cern.ch
[†]  This paper is based on the talk at the 7th International Conference on New Frontiers in Physics (ICNFP 2018), Crete, Greece, 4–12 July 2018.

**Abstract:** The study of open charm meson production provides an efficient tool for detailed investigations of the properties of hot and dense matter formed in nucleus-nucleus collisions. The interpretation of the existing data from the CERN Super Proton Synchrotron (SPS) suffers from a lack of knowledge about the total charm production rate. To overcome this limitation, the heavy-ion program of the NA61/SHINE experiment at the CERN SPS has been upgraded to allow for precise measurements of particles with a short lifetime. A new vertex detector (Small Acceptance version of the Vertex Detector (SAVD)) was constructed to meet the challenges of open charm measurements in nucleus-nucleus collisions. The first exploratory data taking of Pb + Pb collisions at $150A$ GeV/$c$ with the SAVD was performed in 2016, and a $D^0$ signal was extracted in its $D^0 \rightarrow \pi^+ + K^-$ decay channel. This was the first, direct observation of open charm in nucleus-nucleus collisions at the SPS energies. Furthermore, the future plans of open charm measurements in the NA61/SHINE experiment related to the upgraded version of the Vertex Detector are discussed.

**Keywords:** open charm; vertex detector; the NA61/SHINE experiment

## 1. Introduction

One of the important issues related to relativistic heavy-ion collisions is the mechanism of charm production. Several models were developed to describe charm production. Some of them are based on the dynamical approach and some on the statistical approach. The estimates from these approaches for the average number of produced $c$ and $\bar{c}$ pairs ($\langle c\bar{c} \rangle$) for central Pb + Pb collisions at $158A$ GeV/$c$ differ by up to a factor of 50 [1,2]. Therefore, obtaining precise data on $\langle c\bar{c} \rangle$ will allow discriminating between theoretical models and learning about the charm quark and hadron production mechanism. A good estimate of $\langle c\bar{c} \rangle$ can be obtained by measuring production yields of $D^0$, $D^+$, and their antiparticles because these mesons carry about 85% of the total produced charm (this value is predicted by the Parton Hadron String Dynamics (PHSD) model [3]).

The study of open charm meson production is a sensitive tool for detailed investigations of the properties of hot and dense matter formed in nucleus-nucleus collisions at relativistic energies [4–6]. In particular, charm mesons are of special interest in the context of the phase-transition between confined hadronic matter and the Quark-Gluon Plasma (QGP).

The $c\bar{c}$ pairs produced in the collisions are converted into open charm mesons and charmonia (J/$\psi$ mesons and their excited states). The production of charm is expected to be different in confined and deconfined matter. This is caused by different properties of charm carriers in these phases. In confined

matter, the lightest charm carriers are $D$ mesons, whereas in deconfined matter, the lightest carriers are charm quarks. Production of a $D\bar{D}$ pair ($2m_D$ = 3.7 GeV) requires an energy about 1 GeV higher than the production of a the $c\bar{c}$ pair ($2m_c$ = 2.6 GeV). The effective number of degrees of freedom of charm hadrons and charm quarks is similar [7]. Thus, more abundant charm production is expected in deconfined than in confined matter. Consequently, in analogy to strangeness [2,8], a change of the collision energy dependence of the $\langle c\bar{c}\rangle$ yield may be a signal of the onset of deconfinement.

The probability of a $c\bar{c}$ pair hadronizing into a J/$\psi$ meson is defined as $P(c\bar{c} \to J/\psi) \equiv \frac{\langle J/\psi\rangle}{\langle c\bar{c}\rangle} \equiv \frac{\sigma_{J/\psi}}{\sigma_{c\bar{c}}}$. To be able to determine this probability, one needs data on both J/$\psi$ and $c\bar{c}$ yields in the full phase space. This can be done only by studying the charmonium yield relative to the yield of open charm mesons [5].

As has been pointed out by Satz [5], only precise measurement of both the open and hidden charm can provide information on the influence of the final state interaction on charmonia yields in a model independent way.

Up to now, direct measurements of open charm have not been possible at the Super Proton Synchrotron (SPS) energies. The NA38/NA50 and NA60 experiments at the European Organization for Nuclear Research (CERN) provided measurements of J/$\psi$ production at the top SPS energy in the di-muon decay channel [9,10]. In Figure 1 the production of J/$\psi$ in In + In and Pb + Pb collisions is shown as a function of the number of participating nucleons relative to the perturbative Quantum Chromodynamics (pQCD) predictions (assuming normal nuclear absorption in the medium). For a lower number of participants, the yields are consistent with the theoretical estimations. However, at $N_{part} > 200$, the result shows a significant drop, which is known as anomalous J/$\psi$ suppression. This suppression was an important argument for the CERN announcement of the discovery of a new state of matter [11]. Within the Matsui–Satz model [4], the suppression is assumed to be caused by the formation of the QGP. However, due to initial state effects in nucleus-nucleus reactions, like shadowing, parton energy loss, etc. [12], the overall scaled number of the $c\bar{c}$ pairs produced in nuclear collisions may be reduced. Hence, the effect of the medium on J/$\psi$ survival can only be determined by studying the charmonium yield relative to the total charm yield, which can be determined by measuring the yields of open charm mesons [5].

Thus, systematic measurements of open charm production are urgently needed for the interpretation of the existing results on J/$\psi$ suppression. Such measurements would allow disentangling initial and final state effects, revealing the properties of hidden and open charm transport through the dense medium created in nucleus-nucleus collisions.

Figure 2 shows present and future facilities and their region of coverage in the phase diagram of strongly-interacting matter. However, not all of them perform/plan measurements of charm hadrons in nucleus-nucleus collisions:

- The Large Hadron Collider (LHC) and the Relativistic Heavy Ion Collider (RHIC) at high energies ($\sqrt{s_{NN}} \gtrsim 200$ GeV): measurements of charm particles are performed in an acceptance limited to mid-rapidity; this limitation is due to the collider kinematics and related to the detector geometry [13–16]. At very high energy collisions, the multiplicities of $c\bar{c}$ pairs are high, which may lead to the secondary formation of J/$\psi$ mesons, complicating the study of the in-medium effect on J/$\psi$ meson production;
- The RHIC Beam Energy Scan (BES) collider program ($\sqrt{s_{NN}} = 7.7 - 39$ GeV): measurement is not considered in the current program; this may likely be due to difficulties related to collider geometry and kinematics, as well as the low charm production cross-section [17,18];
- The RHIC BES fixed-target program ($\sqrt{s_{NN}} = 3 - 7.7$ GeV): not considered in the current program [19];
- The Nuclotron-based Ion Collider fAcility (NICA) ($\sqrt{s_{NN}} < 11$ GeV): measurements during Stage 2 (after 2023) are considered [20];
- The Japan Proton Accelerator Research Complex Heavy-Ion project (J-PARC-HI) ($\sqrt{s_{NN}} \lesssim 6$ GeV): under consideration; might be possible after 2025 [21];

- The Facility for Antiproton and Ion Research (FAIR) Schwer-Ionen-Synchrotron (SIS-100) ($\sqrt{s_{NN}} \lesssim 5$ GeV): measurements of charm hadrons are not possible due to the very low cross-section at SIS-100, systematic charm measurements are planned with SIS-300 ($\sqrt{s_{NN}} \lesssim 7$ GeV), which is part of the FAIR project; however, no time estimation is available [22].

The conclusion is that only NA61/SHINE will be able to measure open charm production in heavy ion collisions in full phase space in the near future. The corresponding potential measurements at higher (LHC, RHIC) and lower (FAIR, J-PARC) energies are necessary to complement the NA61/SHINE results and establish by collisions the energy dependence of charm production.

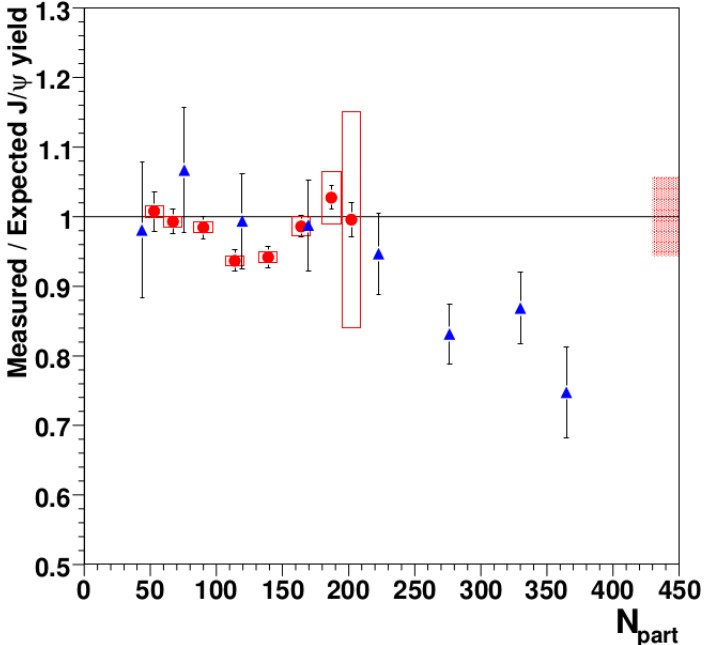

**Figure 1.** $J/\psi$ suppression pattern measured in In + In (red dots) and Pb + Pb (blue triangles) collisions as a function of the number of participants at the top Super Proton Synchrotron (SPS) energy [9,10]. The boxes around the In + In points represent correlated systematic errors. The shaded box on the right shows the uncertainty in the absolute normalization of the $J/\psi$ yields for the In + In interactions.

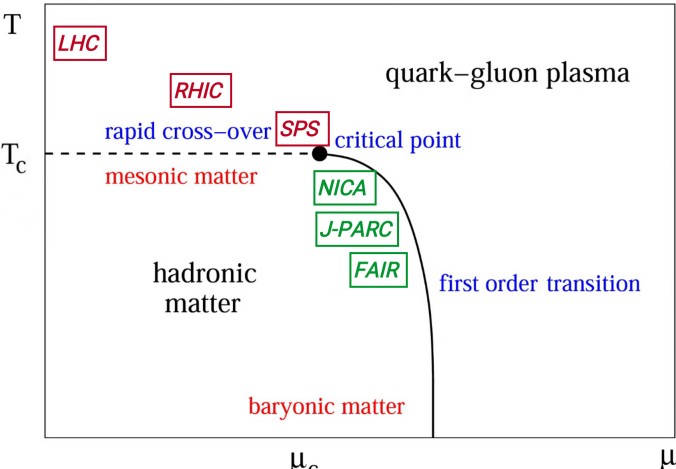

**Figure 2.** Regions in the phase diagram of strongly-interacting matter explored by present (red) and future (green) heavy ion facilities. RHIC, Relativistic Heavy Ion Collider; NICA, Nuclotron-based Ion Collider fAcility; J-PARC, Japan Proton Accelerator Research Complex Heavy-Ion project; FAIR, Facility for Antiproton and Ion Research.

## 2. NA61/SHINE Experiment for Open Charm Measurements

### 2.1. NA61/SHINE Facility

The SPS Heavy Ion and Neutrino Experiment (NA61/SHINE) [23] at CERN was designed for studies of the properties of the onset of deconfinement and the search for the critical point of strongly-interacting matter. These goals are being pursued by investigating p + p, p + A, and A + A collisions at different beam momenta from 13*A*–158*A* GeV/*c* for ions and up to 400 GeV/*c* for protons. It is a fixed target experiment, which is better suited than a typical collider experiment for the detection of strange and multi-strange particles, as well as heavy flavors, like charmed particles. The layout of the experimental setup is shown in Figure 3.

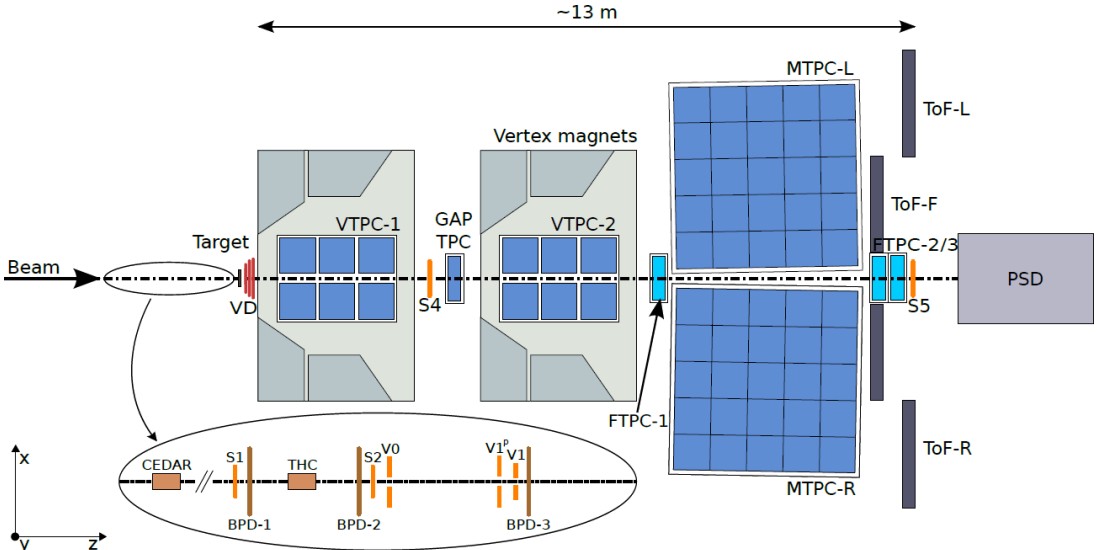

**Figure 3.** Layout of the NA61/SHINE experimental setup (top view, not to scale). VTPC, Vertex Time Projection Chamber; MTPC, Main TPC; PSD, Projectile Spectator Detector; BPD, Beam Position Detector.

The main component of the facility is the large acceptance hadron spectrometer, which was inherited from the NA49 experiment [24]. The setup includes the Beam Position Detectors (BPD), Cherenkov counters, and the scintillator detectors located upstream of the target. They provide information on the timing, charge, and position of beam particles. Charged particle tracking is provided by two Vertex Time Projection Chambers (VTPC-1 and VTPC-2), which are located inside magnets, the gap TPC, and two Main TPCs (MTPC-L on the left side of the beam line and MTPC-R on the right side). These TPCs provide acceptance in the full forward hemisphere, down to $p_T = 0$. The TPCs allow for momentum and charge reconstruction, as well as measurement of mean energy loss per unit path length, which is used for particle identification. The Time-of-Fight (ToF) walls, which are used to improve particle identification, are located behind the main TPCs. The Projectile Spectator Detector (PSD) measures the energy of projectile spectators and delivers information on collision centrality.

Vertex Detector in NA61/SHINE

For open charm measurements in nucleus-nucleus collisions, NA61/SHINE was upgraded with the new Small Acceptance version of the Vertex Detector (SAVD).

As was already mentioned, open charm mesons are difficult to measure because of their low yields and short lifetime. These mesons can in principle be detected in their decay channels into pions and kaons. However, in heavy ion collisions, $\pi$s and $K$s are produced in large numbers by other sources, giving a huge background in the invariant mass distributions (see Figure 4). To distinguish the daughter particles of $D^0$ mesons from hadrons produced at the primary nucleus-nucleus interaction point,

one aims to select only hadron pairs generated in a secondary decay vertex. Vertex reconstruction is done by extrapolating the trajectories back to the target and identifying intersection points. The primary vertex will typically appear as the intersection point of multiple tracks, while the tracks originating from weak decays will intersect at a well-defined displaced point (secondary vertex).

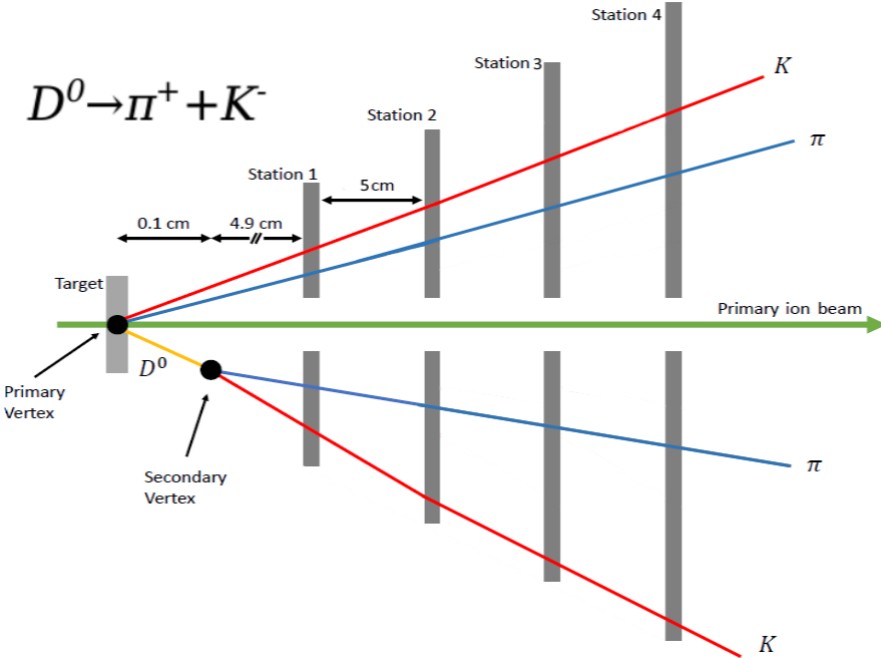

**Figure 4.** Schematics of the reconstruction strategy of the $D^0 \rightarrow \pi^+ + K^-$ decay channel with the help of the vertex detector.

The vertex detector was designed to perform high efficiency tracking and high resolution primary and secondary vertex reconstruction. The detector concept was developed in simulations [25–27], and it only partially covers the total geometrical acceptance of NA61/SHINE. The simulations performed using the A Multi-Phase Transport Model (AMPT) event generator [28] as input show that about 5% of all $D^0 + \overline{D^0}$ in the $\pi$ and $K$ decay channel can be registered by the SAVD and pass background suppression and quality cuts (see the details below).

The SAVD is positioned between the target and the VTPC1 (see Figure 3). It consists of two spectrometer arms (called Jura and Saleve according to the standard NA61/SHINE convention: the Jura arm is located on the Jura Mountains side of the experiment, whereas the Saleve arm is located on the Saleve Mountains side), each composed of four detector planes (stations) of coordinate-sensitive detectors located at a 5, 10, 15, and 20 cm distance from the target. Each arm hosts in total eight sensors (see Figure 5).

High position resolution silicon Minimum Ionizing MOnolithic Active pixel sensors (MIMOSA-26) [29] based on the complementary metal-oxide-semiconductor (CMOS) technology were chosen as the basic detection element of the SAVD stations. The pixel pitch is 18.4 µm in each direction, which leads to a high spatial resolution of 3.5 µm. The $1.06 \times 2.13$ cm$^2$ sensitive area of a single MIMOSA-26 sensor is covered by 1156 columns made of 576 pixels (=663.5k pixels/chip). The sensors have a very low material budget (50 µm thickness), which minimizes the multiple scattering effect in the detection planes. All columns of the sensor are read out in parallel in the rolling shutter mode, and the readout time resolution is 115.2 µs, which is sufficient for data taking at collision rates <1 kHz.

The MIMOSA-26 sensors are mounted on extra light-weight vertical carbon fiber ladders, developed for the upgrade of the Inner Tracking System (ITS) at the A Large Ion Collider Experiment (ALICE) [30]. The ladders are held by C-shaped support frames as illustrated in Figure 3. The ladders have integrated pipes for water cooling. Further, the SAVD and the target were placed in a helium

enclosure (aquarium). The helium helps to minimize the multiple scattering of beam ions and produced particles.

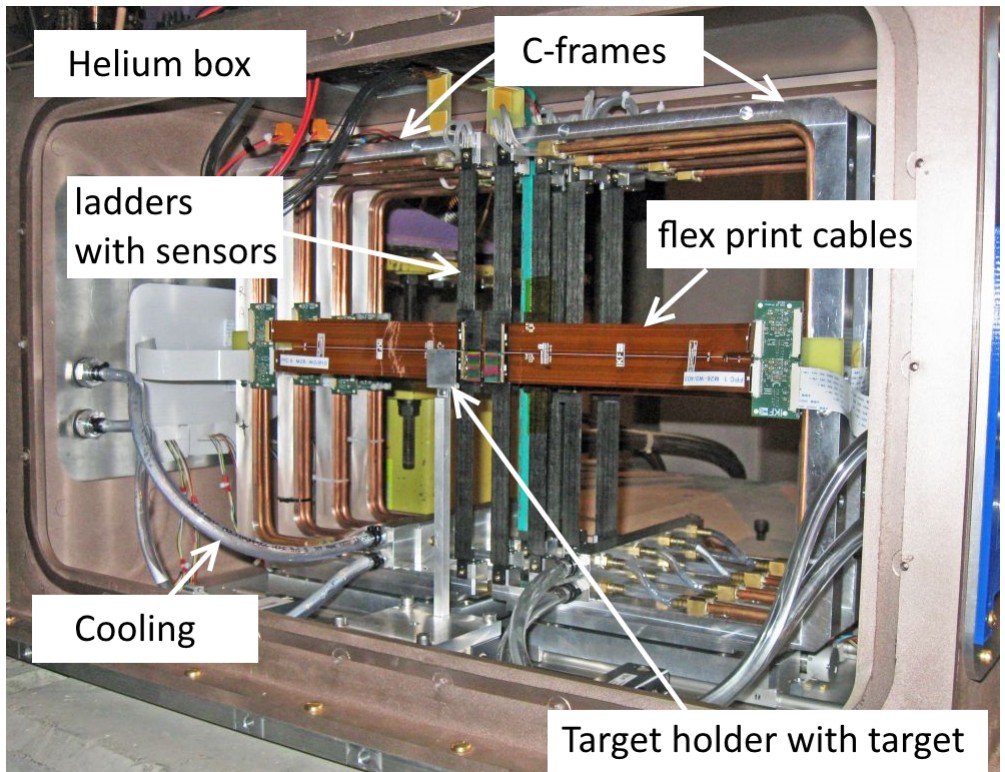

**Figure 5.** Photograph of the Small Acceptance version of the Vertex Detector (SAVD) before closing the detector with the front and exit windows. The detector elements are indicated.

Because the detector is located close to the edge of the VTPC-1 magnet, the magnetic field in the SAVD volume is small and inhomogeneous (0.13–0.25 T)

The sensors are connected to front-end boards that provide sensor biasing. The front-end boards are connected to the data processing Trigger Readout Board (TRB) [31] developed for the updated Micro Vertex Detector (MVD) at the Compressed Baryonic Matter experiment (CBM) [32], which are responsible for the fast and slow control of the sensors. The TRB board has a gigabit-Ethernet interface for data readout.

The MIMOSA-26 sensors are connected to the front-end electronics through 20 cm-long flex print cables. This makes it possible to place the front-end boards outside the VTPC acceptance to reduce the material budget seen by them. Since the MIMOSA-26 sensors do not have internal filters, this made them more susceptible to pick-up noise injected by the long cables. However, adding external capacitor filters reduced the problem.

## 3. Open Charm Measurements with the SAVD

### 3.1. Data Reconstruction in the Vertex Detector

The track finding algorithm is based on a combinatorial method (for pre-tracking) and the Hough transform algorithm (details are discussed in [33]). From the tracking results, it was found that the SAVD spatial resolution is less then 5 µm, as expected.

Primary vertex reconstruction was done by extrapolating the trajectories back to the target and identifying intersection points. The primary vertex will typically appear as the intersection point of multiple tracks, while the tracks originating from secondary decays will intersect at a well-defined displaced point (secondary vertex). A spatial primary vertex resolution of $\sigma_{x,y,z} = 5, 1.5, 30$ µm has

been achieved for test Pb + Pb data taken in 2016. For the Xe + La data, the resolution was found to be $\sigma_{x,y,z} = 1.3, 1.0, 15$ μm. The significant improvement compared to the pilot measurement was due to better setting of the sensor thresholds, resulting in increased pixel efficiency.

The track matching between VD and TPC was done using the following algorithm: at first, tracks were refitted to the VD primary vertex (for primary tracks) or VD clusters from a given station (for secondary tracks) and then interpolated to other VD stations, and the matching clusters were collected. Finally, the whole track was refitted using the Kalman filter algorithm.

### 3.2. Reconstruction of the $D^0 + \overline{D^0}$ Signal

The SAVD tracks matched to TPC tracks were used to search for the $D^0 + \overline{D^0}$ signal. In the current analysis, PID information was not used. Each SAVD track was paired with another SAVD track and was assumed to be either a kaon or pion. Thus, each pair contributed twice in the combinatorial invariant mass distribution. The combinatorial background was several orders of magnitude higher than the $D^0 + \overline{D^0}$ signal due to the low yield of charm particles. In order to reduce the large background, four cuts were applied. The cut values were chosen to maximize the Signal to Noise Ratio (SNR) of the reconstructed $D^0 + \overline{D^0}$ peak and were determined from simulations [26,27]. These cuts were:

- track transverse momentum $p_T > 0.34 \, \text{GeV}/c$;
- track impact parameter $d > 34$ μm;
- longitudinal distance between the $D^0$ decay candidate and the interaction point $V_z > 475$ μm;
- impact parameter D of the back-extrapolated $D^0$ candidate momentum vector D < 21 μm.

Note that the last three cuts were based on information delivered by the SAVD.

## 4. Open Charm Measurements with the SAVD

The SAVD was installed as part of the NA61/SHINE facility and used in 2016 during a Pb + Pb test run at a $150A$ GeV/$c$ beam momentum. An exploratory set of data was collected and analyzed [34]. The main goal of the test was to demonstrate precise tracking in the large track multiplicity environment, the ability of precise vertex reconstruction and the ability to extract the physics result. The obtained primary vertex resolution was $\sigma_x = 5$ μm, $\sigma_y = 1.5$ μm and $\sigma_z = 30$ μm (the difference between $\sigma_x$ and $\sigma_y$ was caused by the presence of a vertical component of the magnetic field in the SAVD volume).

Figure 6 shows the invariant mass distribution of unlike charge daughter candidates with the applied cuts. One observes a peak emerging at 1.82 GeV/$c^2$, which is considered as an indication of $D^0 + \overline{D^0}$ production. The mean value of the peak is shifted relative to the Particle Data Group (PDG) mass value 1.86 GeV/$c^2$ due to the poorly calibrated TPC data. The invariant mass distribution was fitted (red line) using an exponential function to describe the background and a Gaussian to describe the $D^0 + \overline{D^0}$ signal contribution. The indicated errors are statistical only. From the fit, one finds the width of the peak to be $14.6 \pm 3.5$ MeV (which is consistent with the value obtained in simulations including instrumental effects) and the total yield to amount to $62 \pm 19$ with a $\pm 3\sigma$ integrated SNR of 3.3.

The successful performance of the SAVD in 2016 led to the decision to use the device during the Xe + La data taking in 2017. A large statistics dataset was taken for Xe + La interactions at beam momenta of $150A$, $75A$, and $40A$ GeV/$c$. During these measurements, the thresholds of the SAVD sensors were tuned to obtain high hit detection efficiency. This led to significant improvement of vertex resolution precision: $\sigma_x = 1.3$ μm, $\sigma_y = 1.0$ μm, and $\sigma_z = 15$ μm. These data are currently under analysis and are expected to lead to physics results in the coming months.

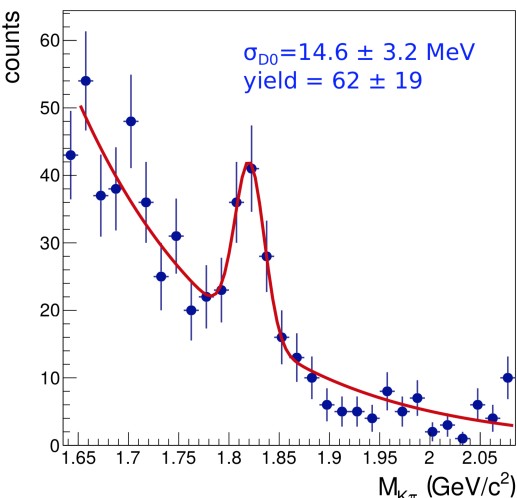

**Figure 6.** Invariant mass distribution of unlike charge sign $\pi, K$ decay track candidates for Pb + Pb collisions at $150A$ GeV/$c$.

## 5. Proposed Open Charm Measurements after CERN Long Shutdown 2

Looking forward, the NA61/SHINE experiment will be upgraded during the CERN long shutdown from 2019–2020 to increase the data taking rate from 80 Hz to 1 kHz [1]. The upgraded VD will be based on the Monolithic Active Pixel sensors (ALPIDE) developed for the ALICE ITS [35] and will have larger acceptance for each station. The proposed program will allow performing systematic studies of $D^0$, $\overline{D}^0$, $D^+$, and $D^-$ production. This study will provide the total $c\bar{c}$ yield in central Pb + Pb collisions needed to investigate the mechanism of charm production in this reaction. Moreover, the data will allow establishing the centrality dependence of $\langle c\bar{c} \rangle$ in Pb + Pb collisions at $150A$ GeV/$c$ and thus address the question of how the formation of QGP impacts $J/\psi$ production.

The simulations performed using A Multi-Phase Transport Model (AMPT) event generator as input show that about 13% (three-times better than for the SAVD) of all $D^0 + \overline{D}^0$ in the $\pi, K$ decay channel can be registered by the upgraded vertex detector and pass background suppression and quality cuts (see Figure 7) and about 9% of all $D^+ + D^-$ in the $2\pi, K$ decay channel (see Figure 8). The total uncertainty of $\langle D^0 \rangle$ and $\langle \overline{D^0} \rangle$ is expected to be about 10% and is dominated by systematic uncertainty.

The data taking plan related to the open charm measurements in 2021–2023 is shown in Table 1. The estimates for open charm yields were made assuming that the mean multiplicity of charm hadrons is proportional to the number of collisions and using yields calculated for central Pb + Pb collisions within the Hadron String Dynamics (HSD) model [36]. In total, 500M minimum bias Pb + Pb events are expected to be collected at top SPS energy. Table 2 lists the expected number of charm mesons in centrality selected Pb + Pb collisions at $150A$ GeV/$c$. From this, it is expected that 76k $D^0$ and $\overline{D}^0$ will be collected, of which 31k will be from the most central collisions. Additionally, 46k $D^+$ and $D^-$ will be collected, of which 19k will be from the most central interactions. These data together with the result for central Pb + Pb collisions at $150A$ GeV/$c$ will start a long-term effort to establish the collision energy dependence of $\langle c\bar{c} \rangle$ and address the question of how the onset of deconfinement impacts charm production.

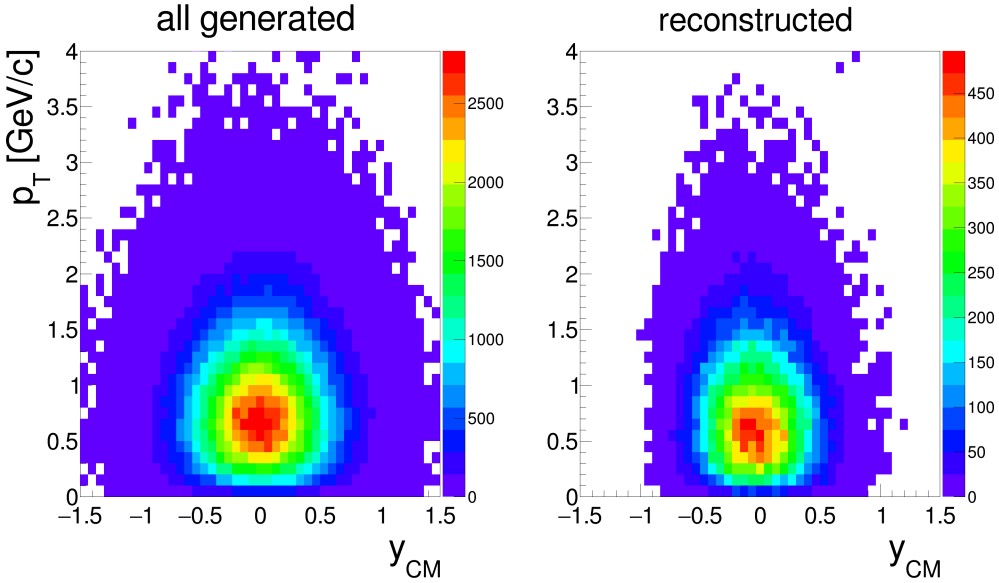

**Figure 7.** A Multi-Phase Transport Model (AMPT) simulation of transverse momentum and rapidity distributions of $D^0 + \overline{D^0}$ mesons produced in central Pb + Pb collisions at $150A$ GeV/$c$ corresponding to 500M recorded events. **Left**: all produced $D^0 + \overline{D^0}$ mesons. **Right**: $D^0 + \overline{D^0}$ mesons in the $\pi, K$ decay channel and both decay products registered by the upgraded VD and the TPCs and passing background suppression and quality cuts.

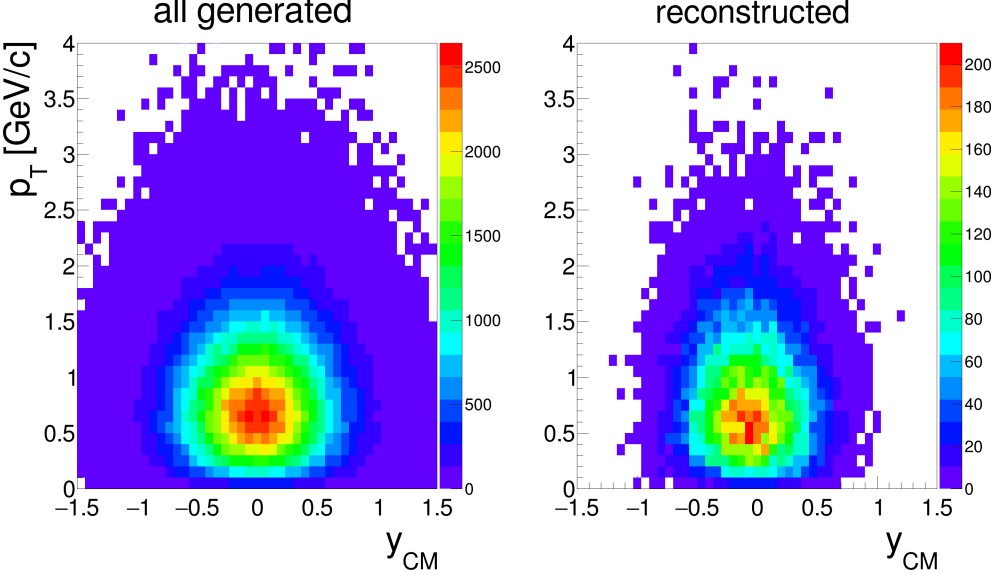

**Figure 8.** AMPT model simulation of transverse momentum and rapidity distributions of $D^+ + D^-$ mesons produced in central Pb + Pb collisions at $150A$ GeV/$c$ corresponding to 500M recorded events. **Left**: all produced $D^+ + D^-$ mesons. **Right**: $D^+ + D^-$ mesons in the $2\pi, K$ decay channel and all decay products registered by the upgraded VD and the TPCs and passing background suppression and quality cuts.

**Table 1.** The NA61/SHINE data taking plan for the open charm measurements.

| Year | Reaction | Events | $D^0 + \overline{D}^0$ | $D^+ + D^-$ |
|------|----------|--------|------------------------|-------------|
| 2021 | Pb + Pb 150$A$ GeV/$c$ | 250M | 38k | 23k |
| 2022 | Pb + Pb 150$A$ GeV/$c$ | 250M | 38k | 23k |
| 2023 | Pb + Pb 40$A$ GeV/$c$ | 250M | 3.6k | 2.1k |

**Table 2.** Expected number of charm mesons in different centrality (0–10%, 10–20%, 20–30%, 30–60%, 60–90%, and 0–90%) selected Pb + Pb collisions at 150*A* GeV/*c* assuming 500M minimum bias events recorded in 2022 and 2023; see the text for detail. The mean number of wounded nucleons $\langle W \rangle$ calculated within the wounded nucleon model is also given, as well as the mean number of binary collisions $\langle N_{COLL} \rangle$.

|  | **0–10%** | **10–20%** | **20–30%** | **30–60%** | **60–90%** | **0–90%** |
|---|---|---|---|---|---|---|
| $D^0 + \overline{D}^0$ | 31k | 20k | 11k | 13k | 1.3k | 76k |
| $D^+ + D^-$ | 19k | 12k | 7k | 8k | 0.8k | 46k |
| $\langle W \rangle$ | 327 | 223 | 159 | 69 | 11 | 105 |
| $\langle N_{COLL} \rangle$ | 769 | 443 | 292 | 102 | 10 | 203 |

## 6. Summary

The SAVD was installed in the NA61/SHINE experiment in 2016 to allow open charm measurements. From the first exploratory data set of Pb + Pb collisions at 150*A* GeV/*c* a $D^0$ signal was extracted in it's $D^0 \rightarrow \pi^+ + K^-$ decay channel. This was the first, direct observation of open charm in nucleus-nucleus collisions at the SPS energies.

In 2017, a large statistic data set has been taken for Xe + La with SAVD at the beam momenta of 150*A*, 75*A* and 40*A* GeV/*c*; and also in 2018 data taking of Pb + Pb collisions at 150*A* GeV/*c* has been performed. This data is now being analysed.

Looking forward, an upgraded version of the Vertex Detector with larger acceptance is being planned. This will allow for starting the detailed research programme of the open charm measurements at SPS energies. Only NA61/SHINE is able to measure open charm production in heavy ion collisions in full phase space and in the beginning of the next decade. These NA61/SHINE results would complement open charm measurements at higher (LHC, RHIC) and lower (FAIR, JPARC) energies.

**Funding:** The work was supported by the Polish National Center for Science Grant 2014/15/B/ST2/02537, the Polish Ministry of Science and Higher Education DSC Grant 7150/E-338/M/2018, No. K/DSC/004993, and the Russian Science Foundation research Grant 16-12-10176.

**Conflicts of Interest:** The authors declare no conflict of interest.

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
