# Peer review of "Open Charm Measurements at CERN SPS Energies with the New Vertex Detector of the NA61/SHINE Experiment: Status and Plans†"

_universe, doi:10.3390/universe5010014_

Reviewer 1 Report

The paper entitled "Open charm measurements at CERN SPS energies with the new Vertex Detector of the NA61/SHINE experiment - status and plans" is written on behalf of the NA61/SHINE Collaboration. The necessity of precise measurements of both open and hidden charm production is explained in the Introduction. Then, the plans of the present and future heavy-ion facility programmes concerning the measurement of charm particles are reviewed. Not all of them are planning to measure charm production, thus outlining the importance of the NA61/SHINE experiment. The main part of the paper is devoted to the Vertex Detector of the NA61/SHINE, its performance, plans for the upgrade and, finally, simulations of its efficiency. 

The paper is clearly written, all important issues are discussed. Therefore, I would like to recommend it for publication.

P.S. Maybe, minor corrections of English and misprints are in order, e.g. 'disCriminate (p.1)",

"...multiplicities ... ARE (instead of IS)... (pp.2-3)", etc. 

Author Response

Thank you for the reviewing my article. I've updated my contribution according to your comments.

Reviewer 2 Report

(1) Please write the full form of the acronyms when you use it for the first time in your article.

(2) I am not so clear about Fig. 2: It is written, at N_part = 200, the results shows a significant drop, but I see the points are statistically consistent. May be saying N_part>200 will be more appropriate. Also what the red boxes and red shaded box at right side of the figure indicate? 

(3) Is it understood why in Fig. 6, the D0 peak is around 1.82 GeV not around the PDG value 1.86 GeV? It will be better to have comments on that.

(4) Table 2: What 0-10%, 10-20% mean? Better to have some explanation. 

Author Response

Thank you for the reviewing my article. I've updated my contribution according to your comments:

(1) I've added the full forms of the used acronyms.

(2) Yes, you are right, better to write >200.The boxes around the In+In points represent correlated systematic errors. The shaded box on the right shows the uncertainty in the absolute normalisation of the J/$\psi$ yields for the In+In interactions. 

(3) Indeed the mean value of the peak is shifted relative to the PDG value. It happened due to not well calibrated TPC data. After calibration of the TPC data would be done, one should expect (and it is like this in other data samples) mean value be the same as in PDG.

(4) In Table 2 values represent different centrality selected classes in collisions.

Reviewer 3 Report

This is a very nice and well written document describing a new vertex detector (SAVD) to allow for precise measurements of short lifetime particles. Though it clearly is a conference report and not the final results,  in fact the authors were clear that physics results would be forthcoming; so this is just very nice status report on their new detector as well as a nice illustration of the statistical precision they will achieve in the future.

The authors both discuss some of the preliminary result as well as discuss the proposed open charm measurements that can be done after the CERN long shutdown. There are other proceedings on this detector that I was able to  quickly find on inspiresHEP, but did seem look like the author has added original material. In particular, the rates after the CERN upgrade seem to be unique to this report, though due to the size of the CERN collaborations, it is hard to due exhaustive searches by hand. The manuscript can be accepted in its present form.

Author Response

Thank you for the reviewing my article.